environmental chemistry/chemical physics

Ce/TiO$_2$, Ho doping, SO$_2$ resistance, low-temperature NH$_3$-SCR

**Author for correspondence:**
Li-min Yan
e-mail: yanlm@shu.edu.cn

This article has been edited by the Royal Society of Chemistry, including the commissioning, peer review process and editorial aspects up to the point of acceptance.

# Enhanced low-temperature NH$_3$-SCR performance of Ce/TiO$_2$ modified by Ho catalyst

## Ting-ting Zhang and Li-min Yan

Shanghai University Microelectronic R&D Center, Shanghai University, Shanghai 200072, People's Republic of China

L-mY, 0000-0001-6777-0075

Holmium was used as a dopant to boost the low-temperature NH$_3$-selective catalytic reduction (SCR) performance of Ce/TiO$_2$ catalyst. It was ascertained that certain amount of Ho-doping species could exceedingly improve the low-temperature SCR activity under 60 000 h$^{-1}$ of Ce/TiO$_2$, accompanied with the improvement of tolerance to H$_2$O and SO$_2$ at 200°C. Characterization results manifested that Ho modification could not only result in inhibiting the growth of TiO$_2$ crystals and the enlargement of specific surface area but also lead to the enhanced redox ability and the increased amount of surface-adsorbed substances, all of which could promote the low-temperature NH$_3$-SCR performance of Ce/TiO$_2$.

## 1. Introduction

Lately, NOx has become one of the significant sources of air pollution. The over-standard concentration of NOx emission was mainly caused by the combustion process of fossil fuel, which has caused many environmental problems such as city smog and pollution [1–4]. Selective catalytic reduction (SCR) is the widely accepted de-NOx technology, and V$_2$O$_5$-WO$_3$ (MoO$_3$)/TiO$_2$ catalyst is the most commercially used catalyst in this SCR system [5]. However, there are still some disadvantages in the SCR system with V$_2$O$_5$-WO$_3$ (MoO$_3$)/TiO$_2$, such as the high operating temperature (300–400°C), the toxicity of vanadium species, low N$_2$ selectivity in the working temperature range [6–9]. Based on these practical disadvantages, it is necessary to study non-vanadium catalysts with better low-temperature SCR performance.

Cerium, one of the most abundant rare-earth metals, has drawn attention due to the high oxygen storage capacity and good redox property. It has been widely applied in catalysis, such as carbon monoxide oxidation and reforming reactions [10–12]. Results of previous research proved that cerium-based oxide catalysts had a good SCR performance. Gao *et al.* [13] reported that Ce/TiO$_2$ by the sol–gel method possesses high

surface area and good redox ability, contributing to its high SCR activity. Vuong *et al.* [14] reported that V/CeTiO$_2$ catalysts showed excellent de-NOx activity at low temperature. Notably, the best one of these V/CeTiO$_2$ catalysts showed almost 100% NO conversion at 190°C. It was also found [15] that doping certain quantity of Ca would increase Ce$^{3+}$ and surface-adsorbed oxygen. Meanwhile, the Brønsted acidity and redox ability were also greatly enhanced. All these factors may be responsible for the enhanced activity. Mosrati [16] recently reported that an impregnated Ce/Ti oxide catalyst with Nb modification presents a 95% NOx conversion at 200°C. Relevant characterization results proved that the Nb introduction decreases the surface area and strengthens the surface acidity. A Ce–Ti oxide catalyst with Cu addition could promote the SO$_2$ resistance of Ce–Ti oxide [17]. Although several catalysts, such as V/CeTiO$_2$ and Ce–Cu–TiO$_2$, have been successfully applied in NH$_3$-SCR, enhanced low-temperature NH$_3$-SCR performance and SO$_2$ resistance of Ce/TiO$_2$ modified by Ho have never been reported.

As a rare earth metal, Ho has been successfully applied for improving the photocatalytic activity of TiO$_2$ [18]. Owing to its electron trap effect of Ho$^{2+}$↔Ho$^{3+}$, the doping of Ho could efficiently enhance the photocatalytic ability of TiO$_2$. Gamal *et al.* [19] reported that the surface of Ho$_2$O$_3$ exposes more Lewis acid sites, which play a vital role in NH$_3$-SCR reaction. It was also reported [20] that Ho-modified Fe–Mn/TiO$_2$ catalyst shows a larger specific area of Fe$_2$O$_3$ phase compared with that of Fe–Mn/TiO$_2$, which results in a board temperature window and high SO$_2$ tolerance in NH$_3$-SCR reaction. However, the investigation of Ce/TiO$_2$ catalyst with Ho addition has not been reported. In this work, Ho is used for improving the low-temperature NH$_3$-SCR activity of Ce/TiO$_2$, and several characterization methods were applied for investigating the promotion mechanism. Furthermore, SO$_2$ + H$_2$O tolerance of the best catalyst was also studied.

# 2. Experimental

## 2.1. Catalyst preparation

The impregnation method was used to prepare the catalysts. Titanium dioxide (anatase, 0.05 mol) was impregnated with cerium nitrate (0.0175 mol) and holmium nitrate in 100 ml deionized water, followed by stirring at 20°C for 3 h. The obtained mixture was dried for 12 h at 100°C and then calcined at 500°C for 4 h. The prepared samples were labelled as Ce$_{0.35}$/TiO$_2$ and Ho$_x$Ce$_{0.35}$/TiO$_2$ (the molar ratios of Ho/Ti and Ce/Ti were $x$ and 0.35, respectively).

## 2.2. Catalyst characterization

Powder X-ray diffraction (XRD) patterns were obtained on a Philips X'pert Pro diffractometer with Ni-filtered Cu K$\alpha$ radiation (0.15408 nm). 2$\theta$ ranged from 10° to 80° with a step size of 0.02°.

The specific surface area was measured by N$_2$ adsorption at −196°C, using an ASAP 2020 volumetric adsorption analyser. Before each precise test, the catalysts were evacuated for 3 h at 300°C. The specific surface area and the pore volume of the samples were calculated by the Brunauer–Emmett–Teller (BET) method and the pore size distributions were derived from the adsorption branches of the isotherms using the Barrett–Joyner–Halenda model.

The H$_2$ temperature-programmed reduction (TPR) experiments were performed on a Micromeritics AutoChem 2920 chemisorption analyser. Typically, 0.1 g sample was pretreated in pure N$_2$ at 400°C for 0.5 h and then cooled to 20°C followed by N$_2$ purging for 0.5 h. The temperature was heated by 10°C min$^{-1}$ from 100 to 800°C in 10 vol% H$_2$/Ar. Thermal conductivity detector monitored H$_2$ consumption in this progress.

The NH$_3$ temperature-programmed desorption (TPD) experiments were carried out on the same equipment as the TPR experiment. As a pretreatment step, 0.1 g samples were purged at 400°C in N$_2$ for 0.5 h and cooled to 30°C. Then the samples were purged in NH$_3$ for 1.0 h. At last, the programmed desorption was carried out at the rate of 10°C min$^{-1}$ (100–500°C) in Ar.

*In situ* diffuse reflectance infrared Fourier transform spectroscopy (DRIFTS) experiments were carried out on a Nicolet 6700 FTIR spectrometer with an MCT/A detector. As a pretreatment step, the catalysts were treated at 450°C in N$_2$ for 0.5 h and cooled to 50°C. Background spectra were recorded in the N$_2$ flow and automatically subtracted from the corresponding spectra. The spectra were recorded by accumulating 64 scans at a 4 cm$^{-1}$ resolution.

## 2.3. Catalytic activity test

SCR activity experiments were performed in a fixed-bed reactor containing 0.4 g catalysts (40–60 mesh) with a GHSV of 60 000 h$^{-1}$. The total gas flow was 200 ml min$^{-1}$, which was premixed in a gas mixer to obtain the simulated gas of [NO] = [NH$_3$] = 500 ppm, [O$_2$] = 3 vol.%, [H$_2$O] = 8 vol.% (when used), [SO$_2$] = 200 ppm (when used) and balanced by N$_2$. Then the mixed gas went into the reactor and the NO and NO$_2$ concentrations were monitored by a 350-XL flue gas analyser. The experiment data were recorded from 100 to 400°C at a steady state. The formulae for NO$_x$ conversion and N$_2$ selectivity were as follows:

$$\text{NO}_x \text{ conversion (\%)} = \frac{[\text{NO}_x]_{\text{in}} - [\text{NO}_x]_{\text{out}}}{[\text{NO}_x]_{\text{in}}} \times 100\% \tag{2.1}$$

$$\text{N}_2 \text{ selectivity} = \left(1 - \frac{2[\text{N}_2\text{O}]_{\text{out}}}{[\text{NOx}]_{\text{in}} + [\text{NH}_3]_{\text{in}} - [\text{NOx}]_{\text{out}} - [\text{NH}_3]_{\text{out}}}\right) \times 100\%. \tag{2.2}$$

Also, NO oxidation conversion was also tested in the same fixed-bed reactor in the same simulated flue gas components without NH$_3$.

# 3. Results and discussion

## 3.1. Catalytic performance

The NOx conversions of various catalysts are plotted as a function of temperature, as exhibited in figure 1a. Among the prepared catalysts, Ce$_{0.35}$/TiO$_2$ and Ho$_{0.35}$/TiO$_2$ showed a limited de-NOx activity (less than 80%) in the entire temperature scope. It is notable that the low-temperature (less than 200°C) catalytic activity of Ce$_{0.35}$/TiO$_2$ was much improved when small amounts of Ho species are doped, as evidenced by the NO conversion of Ho$_{0.15}$Ce$_{0.35}$/TiO$_2$. When the Ho/Ti molar ratio rises to 0.45, the NOx conversion over Ce$_{0.35}$/TiO$_2$ at 150°C was also increased from 22% to 56%. However, further increasing of Ho/Ti molar ratio to 0.6 led to a slight decrease of de-NOx activity in the whole temperature range. Figure 1b shows the N$_2$ selectivity as a function of temperature over Ho$_x$Ce$_{0.35}$/TiO$_2$ catalysts. It could be readily observed that the addition of Ho could enhance the N$_2$ selectivity of Ce$_{0.35}$/TiO$_2$ catalyst. Although all prepared catalysts showed high N$_2$ selectivity in the temperature range of 100–300°C, Ce$_{0.35}$/TiO$_2$ added with Ho exhibited relatively better N$_2$ selectivity above 300°C compared with Ce$_{0.35}$/TiO$_2$ catalyst.

## 3.2. Tolerance of SO$_2$ and H$_2$O

In practical applications, trace amounts of sulfur dioxide and water are still contained in the exhaust gas through the desulfurization unit, which may result in the deactivation of the catalyst. Therefore, the effect of SO$_2$ and water on the SCR activity of the catalyst was studied. Figure 2 depicts the catalytic performance of Ce$_{0.35}$/TiO$_2$ and Ho$_{0.45}$Ce$_{0.35}$/TiO$_2$, as a function of time in the presence of 200 ppm SO$_2$ and 8 vol.% water at 200°C. As exhibited in figure 2, the NO conversion over Ce$_{0.35}$/TiO$_2$ decreased from 52% to 33% after introducing SO$_2$ + H$_2$O for 200 min, then gradually recovered (37%) after the cut off of SO$_2$ + H$_2$O and kept stable during the following test period. By contrast, the presence of SO$_2$ + H$_2$O in the feed gas induced a dramatic decrease of NO conversion over Ho$_x$Ce$_{0.35}$/TiO$_2$ by 10%. After eliminating SO$_2$ + H$_2$O from the feed gas, the conversion of NO over Ho$_x$Ce$_{0.35}$/TiO$_2$ was gradually restored to a certain extent but is less than the initial value (about 72%). All these analyses implied that a better resistance of SO$_2$ + H$_2$O could be achieved by Ho modification.

## 3.3. Brunauer–Emmett–Teller results

BET surface area, total pore volume and average pore were tested. As listed in table 1, the specific surface areas of Ce$_{0.35}$/TiO$_2$, Ho$_{0.35}$/TiO$_2$, Ho$_{0.15}$Ce$_{0.35}$/TiO$_2$, Ho$_{0.3}$Ce$_{0.35}$/TiO$_2$, Ho$_{0.45}$Ce$_{0.35}$/TiO$_2$, and Ho$_{0.6}$Ce$_{0.35}$/TiO$_2$ are 189.61, 157.34, 196.33, 198.34, 204.56 and 203.65 m$^2$ g$^{-1}$, respectively. It is obvious that the specific surface area of Ho$_x$Ce$_{0.35}$/TiO$_2$ became larger as the Ho/Ti molar ratio increased from 0.15 to 0.45. However, doping excess Ho species to Ce/TiO$_2$ (Ho/Ti molar ratio = 0.6) may result in a decrease in BET surface area. Considering the SCR activity results from figure 1a, Ce/

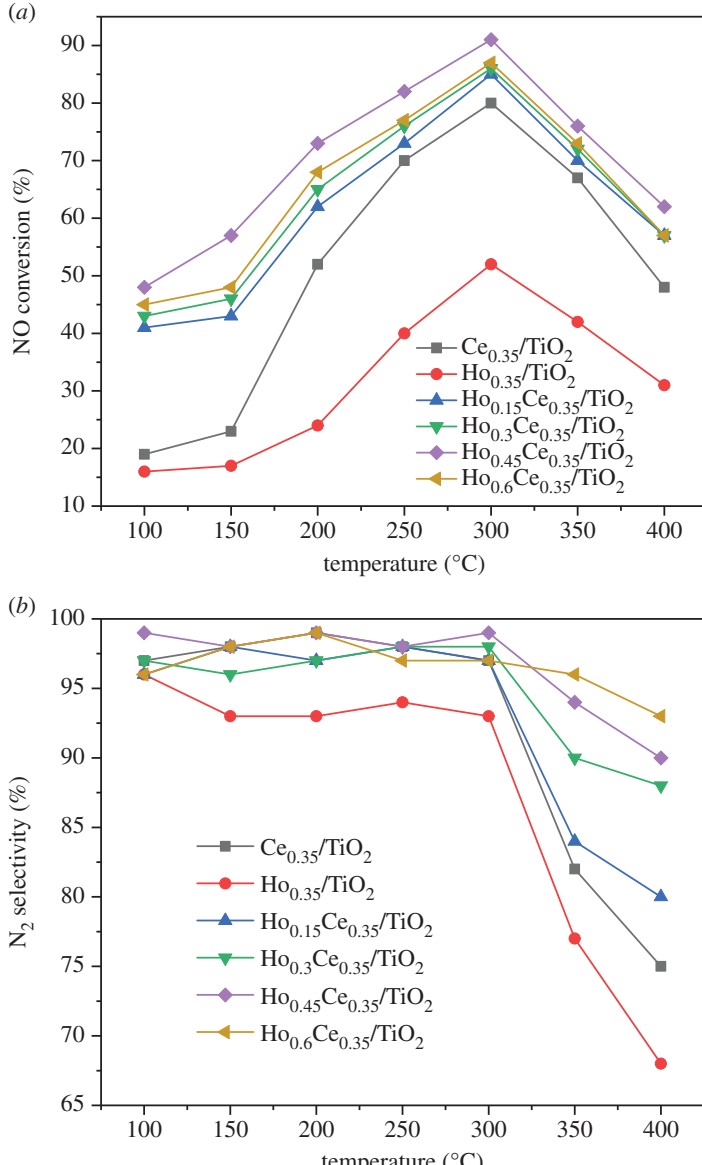

**Figure 1.** (*a*) NOx conversion and (*b*) N$_2$ selectivity in the NH$_3$-SCR reaction over Ho$_x$Ce$_{0.35}$/TiO$_2$ catalysts (500 ppm NO, 500 ppm NH$_3$, 3 vol.% O$_2$, total flow rate 200 ml min$^{-1}$ and GHSV = 60 000 h$^{-1}$).

TiO$_2$ with proper Ho species modification may possess higher active surface area, which is beneficial for the effective contacts with reactants.

## 3.4. Powder X-ray diffraction results

XRD patterns of Ce$_{0.35}$/TiO$_2$ and Ho$_x$Ce$_{0.35}$/TiO$_2$ are shown in figure 3. Only diffraction peaks assigned to TiO$_2$ are detected. Specifically, much anatase-phase TiO$_2$ (PDF-ICDD21-1272) and a little rutile-phase TiO$_2$ (PDF-ICDD21-1276) are observed. A similar phenomenon was also reported by Liu *et al*. [21]. It means that Ce and Ho species are highly dispreading on the surface of TiO$_2$. With the increase of Ho-doping amount, the intensities of all diffraction peaks became weak, suggesting that the introduction of Ho could further reduce the crystallization of TiO$_2$. All of the factors above are favourable to a good SCR performance.

## 3.5. X-ray photoelectron spectroscopy results

Figure 4 exhibits the X-ray photoelectron spectroscopy (XPS) spectra of Ce 3d and O 1s over Ce$_{0.35}$/TiO$_2$ and Ho$_{0.45}$Ce$_{0.35}$/TiO$_2$ catalysts. In addition, the XPS spectrum of Ho 4d over Ho$_{0.45}$Ce$_{0.35}$/TiO$_2$ has been given in figure 4*c*. Table 2 lists the surface element compositions and their chemical states by the XPS technique.

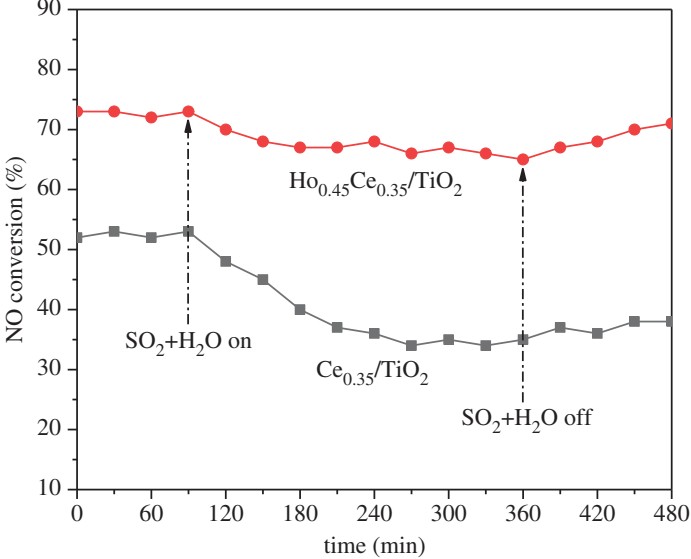

**Figure 2.** NOx conversion of $Ce_{0.35}/TiO_2$ and $Ho_{0.45}Ce_{0.35}/TiO_2$ in the presence of $SO_2$ and $H_2O$ at 200°C (500 ppm NO, 500 ppm $NH_3$, 3 vol.% $O_2$, 8 vol.% $H_2O$, 200 ppm $SO_2$, total flow rate 200 ml min$^{-1}$ and GHSV = 60 000 h$^{-1}$).

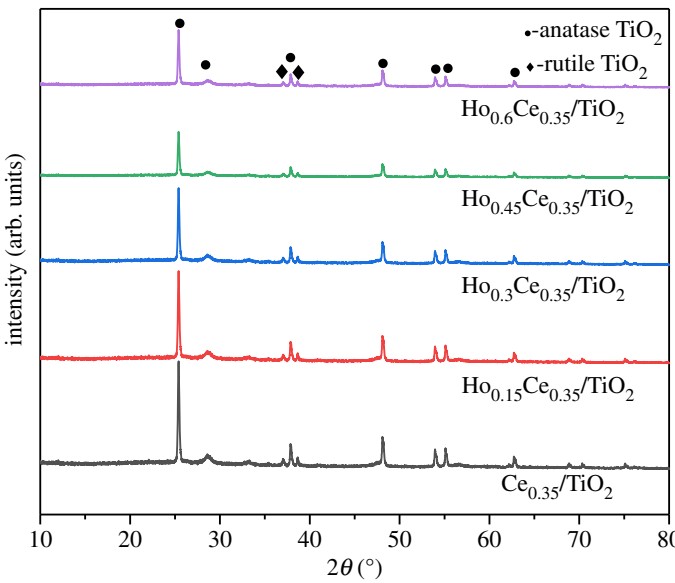

**Figure 3.** XRD patterns of $Ho_xCe_{0.35}/TiO_2$ catalysts.

**Table 1.** Textural parameters of the catalysts.

| samples | BET surface area (m$^2$g$^{-1}$) | pore volume (cm$^3$ g$^{-1}$) | average pore diameter (nm) |
|---|---|---|---|
| $Ce_{0.35}/TiO_2$ | 189.61 | 0.612 | 9.55 |
| $Ho_{0.35}/TiO_2$ | 157.34 | 0.424 | 7.89 |
| $Ho_{0.15}Ce_{0.35}/TiO_2$ | 196.33 | 0.608 | 9.43 |
| $Ho_{0.3}Ce_{0.35}/TiO_2$ | 198.34 | 0.627 | 9.57 |
| $Ho_{0.45}Ce_{0.35}/TiO_2$ | 204.56 | 0.628 | 9.61 |
| $Ho_{0.6}Ce_{0.35}/TiO_2$ | 203.65 | 0.611 | 9.47 |

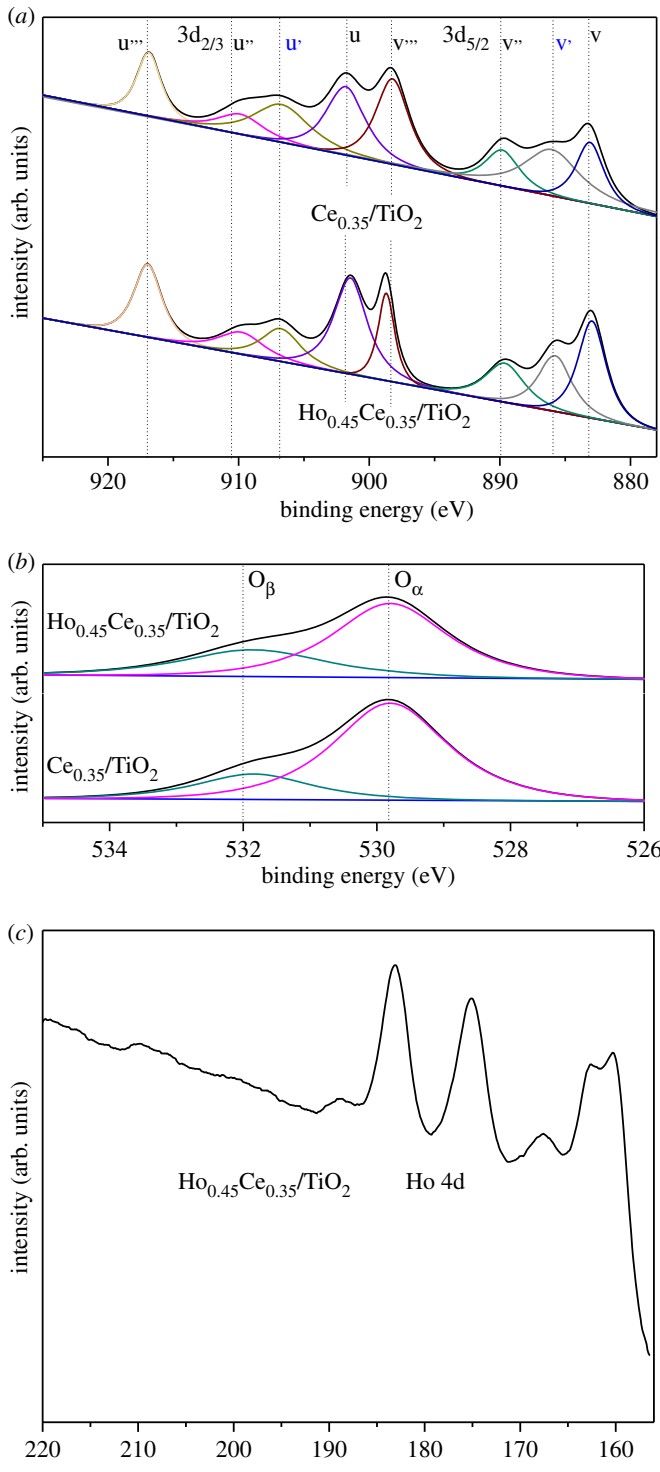

**Figure 4.** XPS spectra of $Ce_{0.35}/TiO_2$ and $Ho_xCe_{0.35}/TiO_2$ catalysts.

As seen in figure 4a, the complicated Ce 3d XPS curves of different samples were made up of eight peaks. u and v peaks belonged to $3d_{3/2}$ and $3d_{5/2}$ spin−orbit components, respectively. u' and v' peaks could be attributed to $Ce^{3+}$ and the other peaks could be assigned to $Ce^{4+}$ [22]. These $Ce^{3+}/Ce^{4+}$ pairs over the catalyst surface were beneficial for not only the storage and release of active oxygen species but also the oxidation of NO to $NO_2$ [23]. Additionally, more $Ce^{3+}$ would promote the generation of more oxygen vacancies, which help to adsorb reactants [24,25]. The factors mentioned above proved to contribute to the progress of the SCR reaction. Thus, it is necessary to study the ratio of $Ce^{3+}/(Ce^{3+} + Ce^{4+})$ over the selected catalysts. The ratio of $Ce^{3+}/(Ce^{3+} + Ce^{4+})$ was calculated according to the

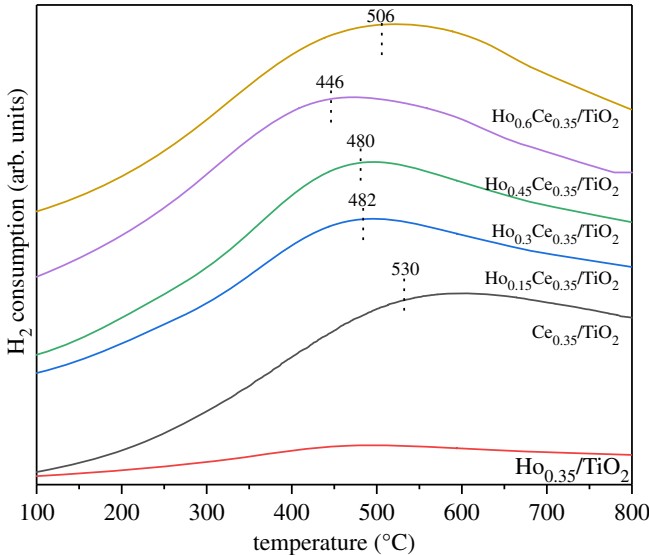

**Figure 5.** $H_2$-TPR patterns of $Ho_xCe_{0.35}/TiO_2$ catalysts.

**Table 2.** Surface elemental analysis by XPS.

| samples | atomic concentration (%) | | | | $Ce^{3+}/(Ce^{3+} + Ce^{4+})$ | $O_\beta/(O_\beta + O_\alpha)$ |
| | Ce | Ti | O | Ho | | |
|---|---|---|---|---|---|---|
| $Ce_{0.35}/TiO_2$ | 4.63 | 13.22 | 82.15 | — | 22.33 | 23.42 |
| $Ho_{0.45}Ce_{0.35}/TiO_2$ | 4.56 | 12.98 | 80.35 | 2.11 | 31.45 | 33.25 |

peak area ratio of the $Ce^{3+}$ and $Ce^{4+}$ peaks. The corresponding results are listed in table 2: $Ho_{0.45}Ce_{0.35}/TiO_2$ (31.45%) and $Ce_{0.35}/TiO_2$ (22.33%). Thus, Ho-doping could promote the transformation of $Ce^{4+}$ to $Ce^{3+}$ over the catalyst surface, which could also effectively improve the SCR activity of $Ce_{0.35}/TiO_2$.

Figure 4b shows that the O 1s XPS spectra of $Ce_{0.35}/TiO_2$ and $Ho_{0.45}Ce_{0.35}/TiO_2$ consisted of two peaks, lattice oxygen (binding energy = 529.8 eV, labelled as $O_\alpha$) and chemisorbed oxygen (binding energy = 532 eV, labelled as $O_\beta$) [26,27]. It is well recognized that $O_\beta$ is more active than $O_\alpha$ in the oxidation reactions of NO to $NO_2$ [28], which is beneficial for the 'fast SCR' reaction ($NO + NO_2 + 2NH_3 = 2N_2 + 3H_2O$). 'Fast SCR' reaction has been proved conducive to the improvement of the low-temperature SCR activity [29]. The $O_\beta/(O_\alpha + O_\beta)$ ratio was calculated and is presented in table 2. It could be observed that $Ho_{0.45}Ce_{0.35}/TiO_2$ has a bigger $O_\beta/(O_\alpha + O_\beta)$ ratio than $Ce_{0.35}/TiO_2$, which meant that chemisorbed oxygen over the catalyst surface of $Ce_{0.35}/TiO_2$ with Ho modification was obviously improved. Considering the results of the SCR activity and Ce 3d XPS, the $O_\beta$ ratio result is corresponding with the $Ce^{3+}$ ratio and SCR activity. It may be concluded that more $Ce^{3+}$ was accompanied by an increment of oxygen vacancies and active oxygen species, which played a positive role in the SCR activity. Finally, the XPS spectrum of Ho 4d over $Ho_{0.45}Ce_{0.35}/TiO_2$ is exhibited in figure 4c.

## 3.6. $H_2$temperature-programmed reduction results

$H_2$-TPR was performed for studying the redox ability of catalysts. In figure 5, no obvious reduction peak of $Ho_{0.35}/TiO_2$ is observed. The reduction peak of $Ce_{0.35}/TiO_2$ at about 530°C belonged to the reduction of $Ce^{4+}$ to $Ce^{3+}$ [30,31]. With the introduction of Ho to $Ce_{0.35}/TiO_2$, the reduction peak of surface $Ce^{4+}$ moved to lower temperature, which could significantly improve the mobility of surface O owing to the strong synergetic effect between Ti, Ce and Ho species. It was also reported that the synergetic effect could lead to the rise of abundant O defects [32,33]. More O defects were beneficial for the improvement of SCR activity because they could promote O diffusion from the subsurface layer and progressively proceed more in-depth into the bulk [34,35]. It could also be observed that $Ho_{0.45}Ce_{0.35}/TiO_2$ showed the lowest reduction temperature at 446°C and this result is corresponding with its best

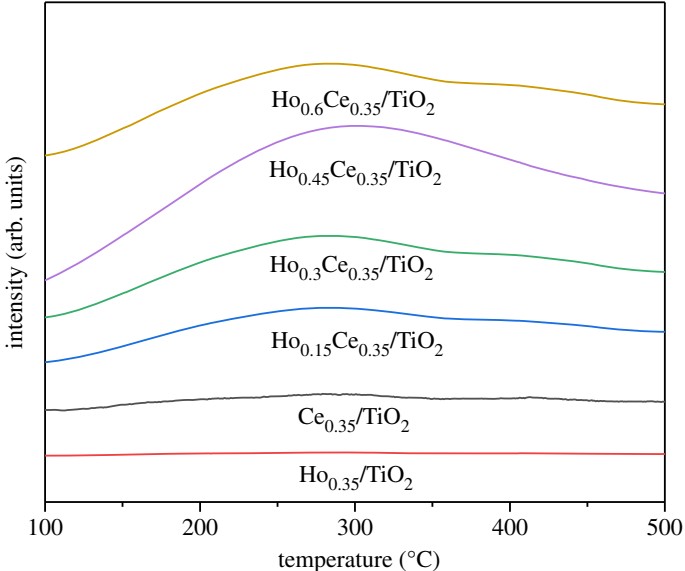

**Figure 6.** NH$_3$-TPD patterns of Ho$_x$Ce$_{0.35}$/TiO$_2$ catalysts.

SCR performance. It seems that further increasing the Ho amount would increase the catalyst reduction temperature. In conclusion, the stronger oxidation–reduction ability of Ho$_{0.45}$Ce$_{0.35}$/TiO$_2$ is beneficial for the outstanding SCR reaction performance.

## 3.7. NH$_3$ temperature-programmed desorption

Figure 6 shows the effect of Ho modification on NH$_3$ desorption behaviour of the prepared samples. From figure 6, no obvious desorption peak of Ho$_{0.35}$/TiO$_2$ was observed and the peak area of Ce$_{0.35}$/TiO$_2$ is shallow. After the introduction of Ho, the peak surface area gradually increases and the NH$_3$-TPD profiles existed as a broad peak with the full range of 120–450°C, which included physically adsorbed NH$_3$, the chemically adsorbed species including adsorbed NH$_3$ species on Brønsted acid sites and strongly adsorbed on Lewis acid sites [33,36,37]. Thus, more surface sites were available on the Ce$_{0.35}$/TiO$_2$ surface for NH$_3$ adsorption after introducing Ho, which could be evidenced by the largest desorption peak area of Ho$_{0.45}$Ce$_{0.35}$/TiO$_2$. The phenomenon could also indicate that Ho$_{0.45}$Ce$_{0.35}$/TiO$_2$ possesses the most potent surface acidity. Thus the adsorption of NH$_3$ over it could be boosted and the SCR activity could be promoted correspondingly.

## 3.8. NO oxidation

Figure 7 exhibits the NO conversion of NO oxidation reaction over the prepared catalysts. It could be easily seen that the NO oxidation conversions over Ce$_{0.35}$/TiO$_2$ and Ho$_{0.35}$/TiO$_2$ are very low (below 25%) during 100–400°C, which is consistent with the lowest SCR activity due to the inefficient conversion from NO to NO$_2$. The activity curves of other catalyst samples demonstrate a parabolic trend, which is an indication of the conversion from the kinetically controlled regime to thermo-dynamically controlled regime [38]. Especially, Ho$_{0.45}$Ce$_{0.35}$/TiO$_2$ has a more significant effect on NO oxidation than other samples. The formation of more NO$_2$ on the catalyst surface facilitates NOx reduction in the low-temperature range, which was also corresponding with the XPS results. Although Ho$_{0.6}$Ce$_{0.35}$/TiO$_2$ had the highest oxidation activity of NO to NO$_2$ of all samples, it exhibited a relatively lower de-NOx activity compared with Ho$_{0.45}$Ce$_{0.35}$/TiO$_2$, which may be attributed to its decreased specific surface area leading to the decreased adsorbed NH$_3$ species.

## 3.9. *In situ* diffuse reflectance infrared Fourier transform spectroscopy results

### 3.9.1. NH$_3$ adsorption

Figure 8a shows the DRIFT spectra of NH$_3$ adsorption over Ce$_{0.35}$/TiO$_2$ at different temperatures. The bands at 1599, 1161 cm$^{-1}$ with a shoulder at 1109 cm$^{-1}$ attributed to the coordinated NH$_3$ linked to

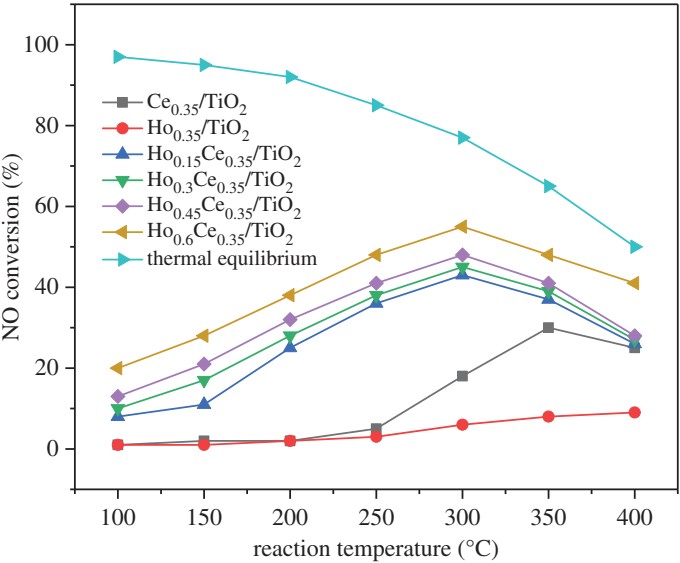

**Figure 7.** Oxidation activity of NO to NO$_2$ by O$_2$ over different catalysts (500 ppm NO, 3 vol.% O$_2$ and 200 ml min$^{-1}$ total flow rate).

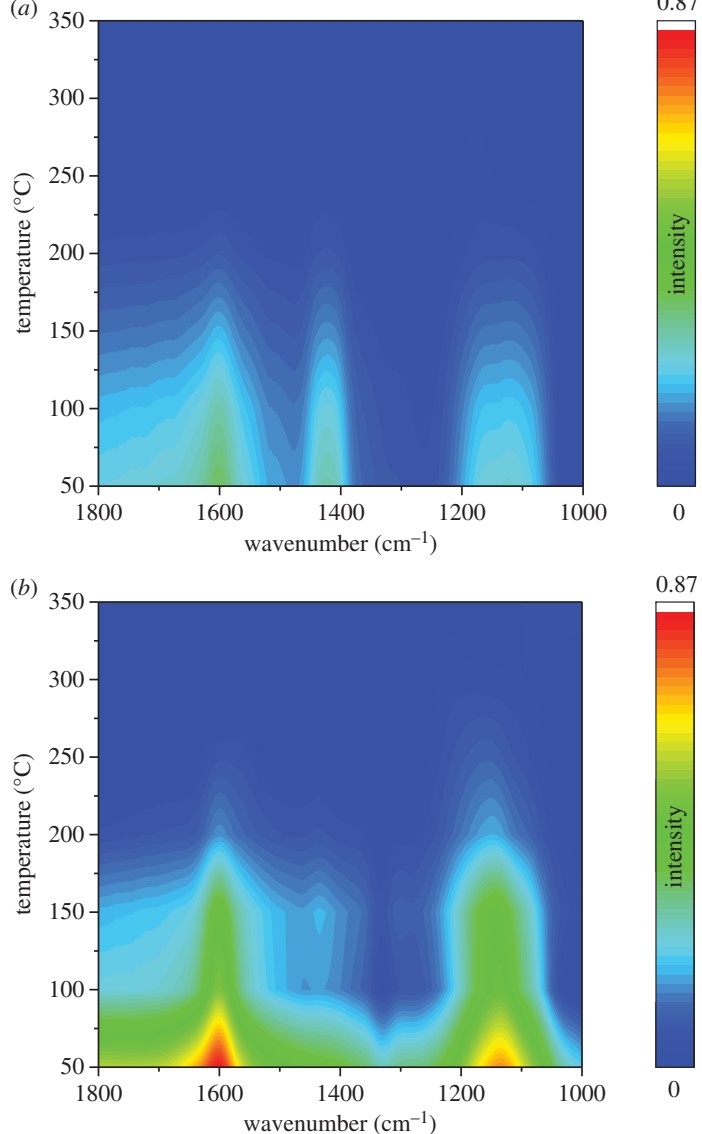

**Figure 8.** *In situ* DRIFTS of NH$_3$ adsorption with increasing temperature from 50 to 350°C: (*a*) Ce$_{0.35}$/TiO$_2$ and (*b*) Ho$_{0.45}$Ce$_{0.35}$/TiO$_2$.

Lewis acid sites (NH$_3$-L) [39,40] could be observed. The band at 1418 cm$^{-1}$ could be assigned to NH$_4^+$ species on Brønsted acid sites (NH$_4^+$-B). Notably, all the bands linked to NH$_3$ species decrease with the temperature increasing owing to the desorption effect.

Figure 8b exhibits the DRIFT spectra of NH$_3$ adsorption over Ho$_x$Ce$_{0.35}$/TiO$_2$. Similar to the spectra over Ho$_x$Ce$_{0.35}$/TiO$_2$, the NH$_3$-L bands (1599 and 1143 cm$^{-1}$) and the NH$_4^+$-B band (1432 cm$^{-1}$) could also be seen. However, the band intensity of adsorbed NH$_3$ over Ho$_x$Ce$_{0.35}$/TiO$_2$ was much stronger than that over Ce$_{0.35}$/TiO$_2$, which indicated that the introduction of Ho species could greatly increase the quantity of both Lewis acid sites and Brønsted acid sites. Previous study by Chen *et al*. [41] and Zhou *et al*. [42] reported that more Brønsted acid sites could help in the generation of adsorbed NH$_3$ species, thus promoting the low-temperature SCR performance. It should also be noted that the intensity of the bands at 1432 cm$^{-1}$ assigned to Brønsted acid sites in figure 8b decreases faster with temperature rising in comparison with those assigned to Lewis acid sites, suggesting NH$_3$ bonded to Lewis acid sites possessed a better thermostability than that bonded to Brønsted acid sites [41].

### 3.9.2. NO + O$_2$ adsorption

Figure 9a shows the DRIFT spectra of NO + O$_2$ adsorption over Ce$_{0.35}$/TiO$_2$ at different temperatures. The bands at 1577, 1536 cm$^{-1}$ attributed to bidentate nitrate could be clearly observed; the band at 1599 cm$^{-1}$ could be assigned to ad-NO$_2$ and the band at 1241 cm$^{-1}$ could be assigned to bridging nitrates [43–45]. It could be observed that all the bands decrease with the temperature increasing owing to the drop in thermal stability.

Figure 9b exhibits the DRIFT spectra of NO + O$_2$ adsorption over Ho$_{0.45}$Ce$_{0.35}$/TiO$_2$ at different temperatures. As shown in figure 9b, the peaks at 1600 cm$^{-1}$ and 1564 cm$^{-1}$ belonged to ad-NO$_2$ and bidentate nitrate. The peak at 1232 cm$^{-1}$ belonged to bridging nitrates [44]. In comparison with that shown in figure 9a, the peak intensity of Ho$_{0.45}$Ce$_{0.35}$/TiO$_2$ was stronger than that of Ce$_{0.35}$/TiO$_2$, which meant that Ho-doping could greatly improve NOx adsorption of Ce$_{0.35}$/TiO$_2$ catalyst.

## 3.10. Promotion mechanism

As evidenced by electronic supplementary material, figure S1, all the adsorbed reactants, including ad-NH$_3$ and ad-NO$_X$ on Ho$_{0.45}$Ce$_{0.35}$/TiO$_2$, could participate in the NH$_3$-SCR reaction. Considering all analysis results given above, doping proper amount of Ho into Ce$_{0.35}$/TiO$_2$ could generate more active NH$_3$ and NO$_x$ species on its surface. After adding Ho species, the generation of more Ce$^{3+}$ and O$_\beta$ over Ho$_{0.45}$Ce$_{0.35}$/TiO$_2$ has a facilitation effect on the conversion from NO to NO$_2$. Thus, the Langmuir–Hinshelwood (L–H) mechanism and Eley–Rideal (E–R) mechanism should be mainly responsible for the promoted low-temperature NH$_3$-SCR activity over Ho$_{0.45}$Ce$_{0.35}$/TiO$_2$, which could be described by the following processes:

(1) L–H mechanism:

$$NO + O_2(g) \rightarrow NO_2(ad) \tag{3.1}$$

$$NH_3(g) \xrightarrow{Ce^{4+}} NH_3(ad) \quad \text{(Lewis acid sites)}. \tag{3.2}$$

'Fast SCR' reaction:

$$NO_2(ad) + 2NH_3(ad) + NO(g) \rightarrow 2N_2(g) + 3H_2O(g) \tag{3.3}$$

$$NH_3(g) \xrightarrow{Ce^{3+}} NH_4^+(ad) \text{ over Brønsted acid sites} \tag{3.4}$$

$$NH_4^+ + e^- + NO_2(ad) \rightarrow NH_4NO_2(ad) \rightarrow N_2 + H_2O. \tag{3.5}$$

(2) E–R mechanism:

$$NH_3(g) \xrightarrow{Ce^{4+}} NH_3(ad) \quad \text{(on Lewis acid sites)} \tag{3.6}$$

$$O_2(g) \rightarrow 2O(ad) \tag{3.7}$$

$$NH_3(ad) + O(ad) \rightarrow NH_2(ad) + OH(ad) \tag{3.8}$$

$$NO(g) + NH_2(ad) \rightarrow NH_2NO(ad) \tag{3.9}$$

$$NH_2NO(ad) \rightarrow N_2(g) + H_2O. \tag{3.10}$$

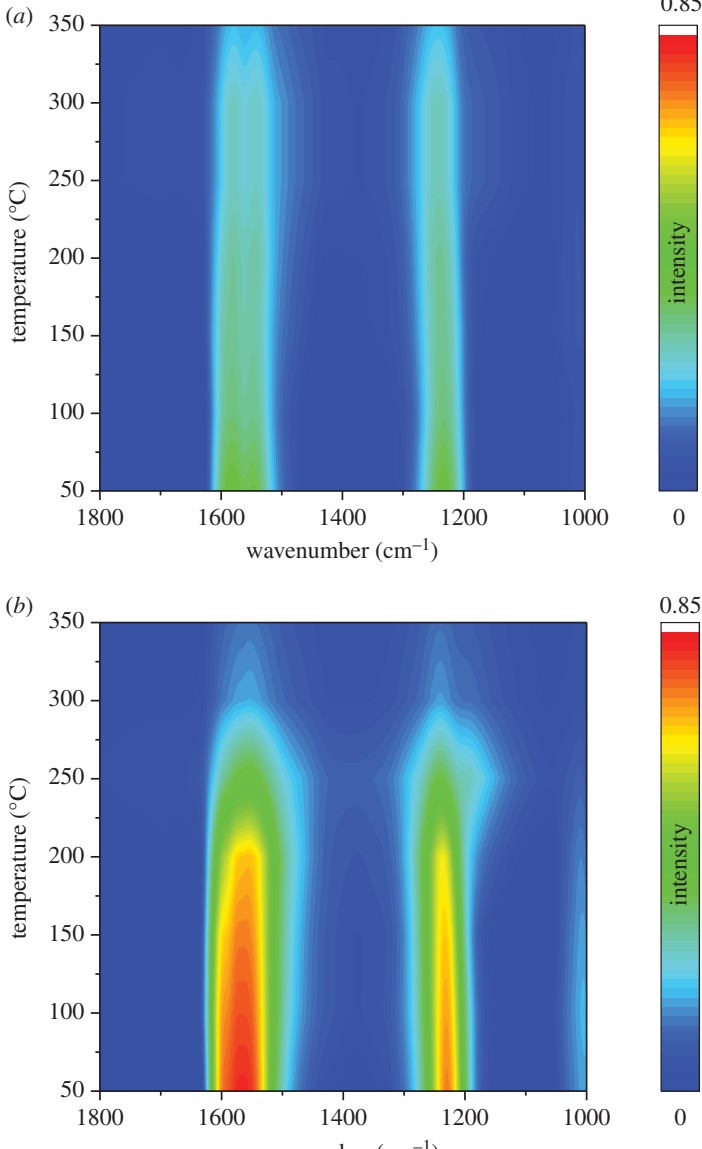

**Figure 9.** *In situ* DRIFTS of NO + O$_2$ adsorption with increasing temperature from 50 to 350°C: (*a*) Ce$_{0.35}$/TiO$_2$ and (*b*) Ho$_{0.45}$Ce$_{0.35}$/TiO$_2$.

## 4. Conclusion

In summary, Ce$_{0.35}$/TiO$_2$ modified with a certain amount of Ho shows an outstanding low-temperature SCR performance and superior SO$_2$ + H$_2$O durability, which could boost the practical application of Ce/TiO$_2$. *In situ* DRIFTS results revealed that the introduction of Ho species could efficiently promote both active ad-NH$_3$ and ad-NOx species on Ce$_{0.35}$/TiO$_2$. Moreover, all of these could contribute to the low-temperature SCR activity of Ce$_{0.35}$Ho$_{0.45}$/TiO$_2$ through L–H route and E–R route.

Ethics. Shanghai University Academic Committee approved the study, and the study was also approved by the National Key Research and Development Program of China (no. 2017YFB0404503). Informed consent for the participants to participate in the study has been received. All authors have been personally and actively involved in substantive work leading to the report, and will hold themselves jointly and individually responsible for its content.

Data accessibility. Our data are deposited at: http://dx.doi.org/10.5061/dryad.c86d5m0 [46].

Authors' contributions. T.-t.Z. designed the study, performed the laboratory experiment and wrote the manuscript. L.-m.Y. assisted in analysing experimental data and editing the manuscript for important intellectual content, and gave the final approval for publication.

Competing interests. We declare we have no competing interests.

Funding. Financial support came from the National Key Research and Development Program of China, no. 2017YFB0404503.

Acknowledgements. We are grateful to reviewers who provided comments that substantially improved the manuscript.

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
