## [Reviewer comments · Royal Society Open Science]

Review History

RSOS-182120.R0 (Original submission)

Review form: Reviewer 1

Is the manuscript scientifically sound in its present form?

Yes

Are the interpretations and conclusions justified by the results?

Yes

Is the language acceptable?

Yes

Is it clear how to access all supporting data?

Yes

Do you have any ethical concerns with this paper?

No

Have you any concerns about statistical analyses in this paper?

No

Recommendation?

Accept with minor revision (please list in comments)

Comments to the Author(s)

In this manuscript, the authors described a study on low-temperature NH₃-SCR performance of Ce/TiO₂ modified by Ho catalyst. A series of Ce/TiO₂ catalysts with different Ho/Ti ratios were examined and compared. The authors designed and performed several experiments such as XRD, H₂-TPR, NH₃-TPD, DRIFTS analysis etc. to evaluate the effect of Ho on the Ce/TiO₂ catalyst. Adequate data and results such as SO₂ and H₂O tolerance, BET, NH₃ adsorption, NO+O₂ adsorption have been presented and discussed by the authors subsequently and appropriate conclusion has been drawn. For example, the Ho modified Ce/TiO₂ catalyst was able to demonstrate higher intensity of NH₃ and NO+O₂ adsorption with increasing temperature in the DRIFTS study which well explains the observation that Holmium could effectively improve the low-temperature SCR activity of Ce/TiO₂. From this work, the readers would be able to get new insights on the properties of Ho modified Ce/TiO₂ catalyst which shows potential applications in various fields. Overall, the manuscript is well presented and meets the requirement as publication on Royal Society Open Science. Therefore, I would recommend this manuscript published after considering the following comments.

In the catalytic performance discussion, the authors presented Figure 1a which demonstrated the curve of NO conversion against temperature. Different Ho modified Ce/TiO₂ catalysts and the unmodified catalyst were compared here. The authors also indicated that when the Ho/Ti molar ratio rise to 0.45, the NO_x conversion over Ce_{0.35}/TiO₂ at 150 °C was increased from 22 % to 56 %. However, further increasing of Ho/Ti molar ratio to 0.6 made a slight decrease of de-NO_x activity in the whole temperature range. Later on, the authors discussed the NO conversion of NO oxidation reaction and presented Figure 7. In Figure 1, the Ho_{0.45}Ce_{0.35}/TiO₂ has shown higher NO conversion than Ho_{0.6}Ce_{0.35}/TiO₂, however, the result is completely opposite in Figure 7. It is recommended to provide a summary of these observations in the discussion to better serve the readers. Additionally, some figure identification numbers used within the manuscript were not correct. Please verify.

Review form: Reviewer 2

Is the manuscript scientifically sound in its present form?

Yes

Are the interpretations and conclusions justified by the results?

Yes

Is the language acceptable?

Yes

Is it clear how to access all supporting data?

No

Do you have any ethical concerns with this paper?

No

Have you any concerns about statistical analyses in this paper?

No

Recommendation?

Accept with minor revision (please list in comments)

Comments to the Author(s)

In this study, Ho was used to modify Ce/TiO₂ catalyst for NH₃-SCR reaction. In my opinion, this manuscript could be accepted for publication after the following revisions:

1. Ho_{0.35}/TiO₂ catalyst should be prepared and tested.
2. The XPS spectra of Ho should be given.
3. E-R mechanism should be taken into consideration.

Decision letter (RSOS-182120.R0)

25-Jan-2019

Dear Dr Yan:

Title: Enhanced low-temperature NH₃-SCR performance of Ce/TiO₂ modified by Ho catalyst
Manuscript ID: RSOS-182120

Thank you for submitting the above manuscript to Royal Society Open Science. On behalf of the Editors and the Royal Society of Chemistry, I am pleased to inform you that your manuscript will be accepted for publication in Royal Society Open Science subject to minor revision in accordance with the referee suggestions. Please find the reviewers' comments at the end of this email.

The reviewers and handling editors have recommended publication, but also suggest some minor revisions to your manuscript. Therefore, I invite you to respond to the comments and revise your manuscript.

Please also include the following statements alongside the other end statements. As we cannot publish your manuscript without these end statements included, if you feel that a given heading is not relevant to your paper, please nevertheless include the heading and explicitly state that it is not relevant to your work. We have included a screenshot example of the end statements for reference.

- Ethics statement

Please clarify whether you received ethical approval from a local ethics committee to carry out your study. If so please include details of this, including the name of the committee that gave consent in a Research Ethics section after your main text. Please also clarify whether you received informed consent for the participants to participate in the study and state this in your Research Ethics section.

OR

Please clarify whether you obtained the necessary licences and approvals from your institutional animal ethics committee before conducting your research. Please provide details of these licences and approvals in an Animal Ethics section after your main text.

OR

Please clarify whether you obtained the appropriate permissions and licences to conduct the fieldwork detailed in your study. Please provide details of these in your methods section.

Because the schedule for publication is very tight, it is a condition of publication that you submit the revised version of your manuscript before 03-Feb-2019. Please note that the revision deadline will expire at 00.00am on this date. If you do not think you will be able to meet this date please let me know immediately.

Best wishes,
Dr Laura Smith
Publishing Editor, Journals

Royal Society of Chemistry
Thomas Graham House

Science Park, Milton Road
Cambridge, CB4 0WF
Royal Society Open Science - Chemistry Editorial Office

RSC Associate Editor:
Comments to the Author:
(There are no comments.)

RSC Subject Editor:
Comments to the Author:
(There are no comments.)

Reviewer comments to Author:
Reviewer: 1

Comments to the Author(s)

In this manuscript, the authors described a study on low-temperature NH₃-SCR performance of Ce/TiO₂ modified by Ho catalyst. A series of Ce/TiO₂ catalysts with different Ho/Ti ratios were examined and compared. The authors designed and performed several experiments such as XRD, H₂-TPR, NH₃-TPD, DRIFTS analysis etc. to evaluate the effect of Ho on the Ce/TiO₂ catalyst. Adequate data and results such as SO₂ and H₂O tolerance, BET, NH₃ adsorption, NO+O₂ adsorption have been presented and discussed by the authors subsequently and appropriate conclusion has been drawn. For example, the Ho modified Ce/TiO₂ catalyst was able to demonstrate higher intensity of NH₃ and NO+O₂ adsorption with increasing temperature in the DRIFTS study which well explains the observation that Holmium could effectively improve the low-temperature SCR activity of Ce/TiO₂. From this work, the readers would be able to get new insights on the properties of Ho modified Ce/TiO₂ catalyst which shows potential applications in various fields. Overall, the manuscript is well presented and meets the requirement as publication on Royal Society Open Science. Therefore, I would recommend this manuscript published after considering the following comments.

In the catalytic performance discussion, the authors presented Figure 1a which demonstrated the curve of NO conversion against temperature. Different Ho modified Ce/TiO₂ catalysts and the unmodified catalyst were compared here. The authors also indicated that when the Ho/Ti molar ratio rise to 0.45, the NO_x conversion over Ce_{0.35}/TiO₂ at 150 °C was increased from 22 % to 56 %. However, further increasing of Ho/Ti molar ratio to 0.6 made a slight decrease of de-NO_x activity in the whole temperature range. Later on, the authors discussed the NO conversion of NO oxidation reaction and presented Figure 7. In Figure 1, the Ho_{0.45}Ce_{0.35}/TiO₂ has shown higher NO conversion than Ho_{0.6}Ce_{0.35}/TiO₂, however, the result is completely opposite in Figure 7. It is recommended to provide a summary of these observations in the discussion to better serve the readers. Additionally, some figure identification numbers used within the manuscript were not correct. Please verify.

Reviewer: 2

Comments to the Author(s)

In this study, Ho was used to modify Ce/TiO₂ catalyst for NH₃-SCR reaction. In my opinion, this manuscript could be accepted for publication after the following revisions:

1. Ho_{0.35}/TiO₂ catalyst should be prepared and tested.
2. The XPS spectra of Ho should be given.
3. E-R mechanism should be taken into consideration.

Author's Response to Decision Letter for (RSOS-182120.R0)

See Appendix A.

Decision letter (RSOS-182120.R1)

06-Feb-2019

Dear Dr Yan:

Title: Enhanced low-temperature NH₃-SCR performance of Ce/TiO₂ modified by Ho catalyst
Manuscript ID: RSOS-182120.R1

It is a pleasure to accept your manuscript in its current form for publication in Royal Society Open Science. The chemistry content of Royal Society Open Science is published in collaboration with the Royal Society of Chemistry.

RSC Associate Editor
Comments to the Author:
(There are no comments.)

Reviewer(s)' Comments to Author:

Appendix A

Detailed response to reviewers

Dear editor:

My manuscript, Enhanced low-temperature NH₃-SCR performance of Ce/TiO₂ modified by Ho catalyst (RSOS-182120), was revised according to your suggestion, and the response is also attached. Many thanks for your suggestion.

Correspondence and phone calls about this paper should be directed to Li-min Yan at the following address, phone and e-mail:

Address: Shanghai University Microelectronic R&D Center, Shanghai University, Shanghai, 200072, P. R. China

Phone: +86-021-56331272

Email: yanlm@shu.edu.cn

Once again, thank you for your kind help to our paper processing.

Sincerely yours:

Li-min Yan

Response to editor:

1. Please also include the following statements alongside the other end statements.

As we cannot publish your manuscript without these end statements included, if

you feel that a given heading is not relevant to your paper, please nevertheless include the heading and explicitly state that it is not relevant to your work.

Response: Ethics have been added in the revised manuscript.

1) A text file of the manuscript (tex, txt, rtf, docx or doc), references, tables (including captions) and figure captions. Do not upload a PDF as your "Main Document".

2) A separate electronic file of each figure (EPS or print-quality PDF preferred (either format should be produced directly from original creation package), or original software format)

3) Included a 100 word media summary of your paper when requested at submission. Please ensure you have entered correct contact details (email, institution and telephone) in your user account

4) Included the raw data to support the claims made in your paper. You can either include your data as electronic supplementary material or upload to a repository and include the relevant doi within your manuscript

5) All supplementary materials accompanying an accepted article will be treated as in their final form. Note that the Royal Society will neither edit nor typeset supplementary material and it will be hosted as provided. Please ensure that the

supplementary material includes the paper details where possible (authors, article title, journal name).

Response: The above requirements have been confirmed.

Response to reviewer 1:

1. However, further increasing of Ho/Ti molar ratio to 0.6 made a slight decrease of de-NO_x activity in the whole temperature range. Later on, the authors discussed the NO conversion of NO oxidation reaction and presented Figure 7. In Figure 1, the Ho_{0.45}Ce_{0.35}/TiO₂ has shown higher NO conversion than Ho_{0.6}Ce_{0.35}/TiO₂, however, the result is completely opposite in Figure 7. It is recommended to provide a summary of these observations in the discussion to better serve the readers.

Response: a summary of these observations in the discussion has been provided in the revised manuscript. Although Ho_{0.6}Ce_{0.35}/TiO₂ possessed the highest oxidation activity of NO to NO₂ in all samples, it exhibited a relatively lower de-NO_x activity compared with Ho_{0.45}Ce_{0.35}/TiO₂, which may attribute to its decreased specific surface area leading to the decreased adsorbed NH₃ species.

2. Additionally, some figure identification numbers used within the manuscript were not correct. Please verify.

Response: some incorrect figure identification numbers used within the manuscript has been verified in the revised manuscript.

Response to reviewer 2:

1. $\text{Ho}_{0.35}/\text{TiO}_2$ catalyst should be prepared and tested.

Response: $\text{Ho}_{0.35}/\text{TiO}_2$ catalyst has be prepared and its NO_x conversion has been tested. Some characterization data for $\text{Ho}_{0.35}/\text{TiO}_2$ catalyst including N_2 selectivity, BET, H_2 -TPR, NH_3 -TPD, NO oxidation have also been added in the revised manuscript.

2. The XPS spectra of Ho should be given.

Response: The XPS spectra of Ho have be given in Fig. 4 (c).

3. E-R mechanism should be taken into consideration.

Response: E-R mechanism has be taken into consideration in the section 3.9.

Finally, thanks again to the editors and reviewers for giving me the opportunity to publish my paper in Royal Society Open Science.